

# A Global Analysis of Dust Diurnal Variability Using CATS Observations

Yan Yu[1], Olga V. Kalashnikova[2], Michael J. Garay[2], Huikyo Lee[2], Myungje Choi[2], Gregory S. Okin[3], John E. Yorks[4], James R. Campbell[5], and Jared Marquis[6]

[1]Atmospheric and Oceanic Sciences Program, Princeton University, Princeton, NJ, USA
[2]Jet Propulsion Laboratory, California Institute of Technology, Pasadena, CA, USA
[3]Department of Geography, University of California, Los Angeles, CA, USA
[4]NASA Goddard Space Flight Center, Greenbelt, MD, USA
[5]Naval Research Laboratory, Monterey, CA, USA
[6]Department of Atmospheric Sciences, University of North Dakota, Grand Forks, ND, USA

*Correspondence to*: Yan Yu (yuyan06@gmail.com)

**Abstract.** The current study investigates the diurnal cycle of dust loading across the global tropics, sub-tropics, and mid-latitudes by analyzing aerosol extinction and typing profiles observed by the Cloud-Aerosol Transport System (CATS) lidar aboard the International Space Station. According to the comparison with ground-based and other satellite observations,

CATS aerosol and dust loading observations exhibits reasonable quality but significant day-night inconsistency. To account for this day-night inconsistency in CATS data quality, the diurnal variability in dust characteristics are currently examined separately for daytime and nighttime periods. Based on an analysis of variance analytical framework, pronounced diurnal variations in dust loading are generally uncovered during daytime periods and over terrestrial areas. The current study identifies statistically significant diurnal variability in dust loading over key dust sources, including the Bodélé Depression,

the West African El Djouf, Rub-al Khali Desert, and western and southern North America, confirming the previous observation-based findings regarding the diurnal cycle of dust emission and underlying meteorological processes in these regions. Significant seasonal dust diurnal variability is identified over the Iraqi and Thar deserts. The identified significant diurnal cycles in dust loading over the rainforests in Amazon and tropical southern Africa are hypothesized to be driven by enhanced dust emission due to wildfires.

## 1 Introduction

Dust mobilization and concentration exhibit substantial diurnal variability around the globe (Knippertz and Stuut, 2014), contributing to the radiative (DeMott et al., 2010; Tegen and Lacis, 1996), biogeochemical (Okin et al., 2004), and societal (Al-Hurban and Al-Ostad, 2010; Furman, 2003) impacts of mineral dust on the Earth system. For example, accurate representation of the diurnal variability in dust loading is the key for realistic simulation of the dust radiative effects on the

surface (Miller et al., 2004; Osipov et al., 2015; Yue et al., 2009). However, current state-of-the-science models continue to struggle with their representation of the dust diurnal cycle. Based on model simulations using different meteorological drivers and dust source parameterizations, Luo et al. (2004) concluded that ~35-70% of the variance of dust mobilization is





associated with diurnal variability of dust mobilization in the world's major dust source regions. But the simulated diurnal variations in dust mobilization and concentration were highly sensitive to the choice of the meteorological driver dataset and

dust source parameterization, suggesting substantial uncertainty in the model-based assessment of dust diurnal variability (e.g., Miller et al., 2004; Yue et al., 2009). Coupled climate or Earth System models are widely used to study the mobilization, transport, and radiative effects of dust aerosols. However, due to the coarse spatial and temporal resolution of these models, the capability of them at accurately capturing the diurnal cycle of near-surface winds or convective systems is largely limited, leading to often incorrect simulation of diurnal cycle in dust emission and concentration (Marsham et al.,

2011; Todd et al., 2008). Given these uncertainties in the simulated global dust diurnal variability, observational characterization of the variations in dust mobilization and concentration provide a valuable benchmark for evaluating and constraining such model simulations.

In addition, an improved understanding of dust diurnal cycle is beneficial to biological and geological studies of desert surfaces as land surface observations are most valuable during low atmospheric aerosol activity. For example, The Earth

Surface Mineral Dust Source Investigation, EMIT, is planned to operate from the International Space Station starting in 2021 to determine the mineral composition of the arid land dust source regions of the Earth. The EMIT space-borne mission will acquire, validate, and deliver updates on the surface mineralogy used to initialize Earth System Models through the use of imaging spectroscopy in the visible to short wavelength infrared (VSWIR) portion of the spectrum. The mission will measure the arid land mineral dust source regions of the Earth, recording the distinct spectral features of the iron oxide,

sulfate, clay, and carbonate minerals on the surface. However, clear-sky conditions are an important requirement for the mission success as the wavelength range of EMIT is sensitive to presence of atmospheric aerosols. EMIT's atmospheric correction team is working on an evaluation of dust temporal variability to optimize observational opportunities of unobscured desert surfaces for EMIT target areas. EMIT requirements partially motivated this study.

The diurnal cycle in dust mobilization has been documented for several major dust sources and associated with various

meteorological processes. For example, over the Bodélé Depression, the leading dust source in the globe (Engelstaedter et al., 2006; Kocha et al., 2013; Koren et al., 2006; N'Tchayi Mbourou et al., 1997; Washington and Todd, 2005), dust mobilization is predominantly driven by high surface wind speeds that peak in the morning with the breakdown of the nocturnal low-level jet (Wagner et al., 2016). Over the Western African El Djouf, the second largest dust source in North Africa (Yu et al., 2018), a great portion of dust storms are caused by strong downbursts associated with deep convection in

the afternoon (Fiedler et al., 2013; Heinold et al., 2013). Over the Iraqi Desert in the Middle East, summertime dust activation is primarily driven by the strong, persistent Shamal wind, which peaks around local noon with intensified low-level temperature gradient (Yu et al., 2016). Beyond these studies that focused on a specific dust source region, there has been limited global analysis of the observed diurnal variability in dust mobilization and concentration, with the exception of some modelling studies (e.g. Yue et al., 2009).

Satellite- and ground-based aerosol loading measurements and aerosol type classifications are useful for quantitative assessment of the observed global dust diurnal variability, but several methodological challenges have to first be addressed.



Sun-synchronous, passive satellite instruments, such as the Multi-angle Imaging SpectroRadiometer (MISR) (Diner et al., 1998; Kalashnikova et al., 2005) on the Terra satellite and the Moderate Resolution Imaging Spectroradiometer (MODIS) on both the Terra and Aqua satellites, provide observations of dust aerosol optical depth (DAOD), but only cover several

snapshots during the daytime (e.g. Kocha et al., 2013). Observations from lidar instruments, such as the Cloud-Aerosol Lidar with Orthogonal Polarization (CALIOP) on the polar-orbiting Cloud-Aerosol Lidar and Infrared Pathfinder Satellite Observation (CALIPSO) satellite (Winker et al., 2009), provide both daytime and nighttime measurements of vertically resolved aerosol extinction and aerosol type information. However, CALIOP only samples at most two, fixed temporal points at each location on the Earth, and is therefore insufficient for studying the full diurnal cycle (e.g. Kocha et al., 2013).

Geostationary sensors, such as the Advanced Himawari Imager on the Himawari-8 and -9 satellites (Bessho et al., 2016), the Geostationary Ocean Color Imager on the Communication, Ocean and Meteorological Satellite (Choi et al., 2018), the Advanced Baseline Imager on the GOES-16/17 satellites (Schmit et al., 2017), and the Spinning Enhanced Visible and Infrared Imager (SEVIRI) instrument aboard the Meteosat Second Generation satellite (Schepanski et al., 2007), only provide observations over a certain region. AErosol RObotic NETwork (AERONET) (Holben et al., 1998) sun photometers

provide hourly or sub-hourly measurements of total column AOD and retrievals of aerosol properties, but have limited global coverage (Giles et al., 2019).

The aforementioned challenges in the observational assessment of global dust diurnal variability may be partly addressed by the Cloud-Aerosol Transport System (CATS) lidar aboard the International Space Station (ISS) (McGill et al., 2015). CATS is an elastic backscatter lidar that operated on the ISS for 33 months from February 2015 to October 2017. The

51˚ inclination of the ISS orbit results in CATS measurements at different local times every overpass, with full diurnal coverage for a given location within a 60-day period (Yorks et al., 2016). By comparing CATS-derived AOD and aerosol vertical distributions with aerosol properties derived from other ground- and satellite-based observations such as AERONET, MODIS, and CALIOP, Lee et al. (2019) found reasonable agreements between aerosol observations from CATS and other sensors, thereby verifying that CATS provides an encouraging opportunity for studying aerosol diurnal variability. By

examining CATS aerosol observations, Lee et al. (2019) further identified strong diurnal cycles in total AOD over North Africa, India, and the Middle East, likely attributed to the diurnal variations in dust generation. Several limitations in this initial study were noted by the authors. First, although they reported a substantially better agreement between CALIOP and CATS AODs during nighttime, likely due to enhanced solar contamination during daytime that affects the data quality of both instruments (Campbell et al., 2012; Pauly et al., 2019), this day-night data inconsistency was not accounted for in the

assessment of aerosol diurnal variability. Furthermore, Lee et al. (2019) did not perform a formal significance test of the aerosol diurnal cycle, leading to a potential overinterpretation over sparsely sampled regions. Therefore, this initial global assessment of aerosol diurnal variability using CATS observations motivates a more sophisticated investigation that, first, accounts for potential day-night data inconsistency and, second, explicitly quantifies the significance of dust diurnal variability.



The current study investigates the observed diurnal variability in dust loading over the global tropics, sub-tropics, and mid-latitudes by examining aerosol extinction and aerosol type observations from CATS. The day-night data quality consistency is assessed through a comparison of daytime and night-time CATS observations with AERONET sun and lunar data. The spatial variations in CATS-derived DAOD is assured with MISR non-spherical, dust AOD and CALIOP DAOD. A statistical approach is undertaken to systematically determine the statistical significance of the dust diurnal variability. The

dust diurnal cycle over key regions are discussed along with the driving meteorological processes. The methods, results, and conclusions/discussion are provided in Sections 2, 3, and 4, respectively.

## 2 Data and Methods

### 2.1 CATS

       CATS Level 2 (L2) Version 3-00 5 km Aerosol Profile products (L20_D-M7.2-V3-00_5kmPro, L20_N-M7.2-V3-

00_5kmPro) are used in the current study for the entire period of CATS operation on the ISS during February 2015 - October 2017. CATS L2 profile data is provided with 5 km horizontal resolution along track, in 533 vertical levels at 60 m vertical resolution, and at a wavelength of 1064 nm (Pauly et al., 2019). Data at 532 nm is also provided by CATS but not recommended for use due to a laser-stabilization issue (Yorks et al., 2016). Thus, only 1064 nm products are used in the current study. The accuracy of the extinction coefficient has improved from Version 2 to Version 3 CATS products, as a

result of several improvements in the retrieval algorithms, especially during the daytime. For aerosol typing, CATS uses layer-integrated 1064 nm depolarization ratio, layer base altitudes and thickness, surface type, and GEOS modeled aerosol species to discriminate dust, smoke, polluted continental, marine, and upper stratosphere/lower troposphere aerosols. In particular, the depolarization ratio represents particle shape and is often used to separate dust from other aerosols. Larger depolarization ratio indicates higher likelihood of dust, because of its nonspherical shape (Vaughan et al., 2009). According

to the comparison with other satellite images and modelling results, CATS successfully captured a large plume of Saharan dust transporting across the Atlantic Ocean on June 17, 2015 (NASA Goddard Space Flight Center, 2017). This plume was elevated to about 6 km above sea level off the African coast and mixed with marine aerosols over the tropical Atlantic Ocean (Supplemental Figure 1).

       CATS data are quality-assured (QA), mainly following (Lee et al., 2019). QA thresholds include: 1)

Extinction_QC_Flag_1064_Fore_FOV = 0, which indicates a non-opaque layer, 2) Feature_Type_Fore_FOV = 3, which indicates pure aerosol, 3) -10 <= Feature_Type_Score_FOV <= -2, which indicates aerosol by negative scores and higher confidence by higher absolute scores, and 4) Extinction_Coefficient_Uncertainty_1064_Fore_FOV <= 10 $km^{-1}$.

       In the current study, both total AOD, as reported in the CATS standard product, and DAOD, as computed from vertical profiles of extinction coefficient and feature/aerosol type, are analyzed. AOD from CATS is compared with AERONET to

assess the data quality during both daytime and nighttime. DAOD is defined here as the vertical integral of aerosol extinction coefficient over "dust" (Aerosol Type = 3) or "dust mixture" (Aerosol Type = 4, which represents the mixture of dust with





other aerosols, such as biomass burning and marine aerosols) pixels, thereby reflecting the total dust loading at each location. The spatial variations in DAOD is evaluated against MISR and CALIOP. In light of the larger uncertainty associated with a reported AOD of zero (Toth et al., 2018), any AOD or DAOD that equal to zero is ignored in the current analysis, following

Campbell et al. (2012). This approach results in ~1000 of DAOD retrievals in any three-hour local time window at each 2˚x2˚ grid cell over the dust source regions, such as North Africa, and fewer than 100 retrievals over remote oceans (Supplemental Figure 2).

## 2.2 AERONET

In order to assess the CATS data quality at both day and night, Version 3 AERONET AOD from both sun and lunar
photometers are analyzed. The level 2 (cloud screened and quality assured) daytime (Giles et al., 2019) and level 1.5 (cloud screened) nighttime AOD observations (Barreto et al., 2019, 2016) at the 1020 nm spectrum band are compared with collocated CATS AOD at 1064 nm. Here a "collocated observation" is identified when the CATS orbit passed anywhere in the ± 0.5 latitude/longitude box of a specific AERONET site within ± 0.5 hour of the corresponding AERONET site observation. Previous observational studies have identified a threshold of 40 km and 3 hours beyond which the spatial and
temporal autocorrelation drops below 80% (Anderson et al., 2003; Omar et al., 2013). In the current study, the relatively broad criteria on spatial and temporal collocation is expected to maintain the spatial and temporal autocorrelation while provide reasonable number of collocated observations, especially for the nighttime comparison (Figure 1). Note that one AERONET measurement is often associated with multiple CATS retrievals in both space and time. In this case, CATS data is averaged spatially and temporally, resulting in only one pair of collocated and averaged CATS observations for a given
collocated incident at each AERONET site.

## 2.3 Comparison with MISR and CALIOP

In order to assess the validity of the spatial and seasonal distribution of CATS DAOD, Version 23, Level 3 MISR nonspherical, dust AOD at 550 nm during the CATS operation period is analyzed here. The MISR nonspherical AOD fraction is often referred to as "fraction of total AOD due to dust," as dust is the primary nonspherical aerosol particle in the
atmosphere, especially over desert regions (Kalashnikova et al., 2005). In light of the narrow swath of MISR and the resulting limited number of collocated observations from CATS and MISR, here global maps of seasonal average DAOD from CATS and MISR are compared, thereby verifying the spatial variations in CATS DAOD. Given the Terra overpass time around 10:30 am local time, only morning data from CATS (8 am to local noon) are used for the seasonal average. In order to achieve sufficient sampling from CATS, DAOD from both CATS and MISR are aggregated into a 2˚latitude x 2˚
longitude grid. Given the different wavelengths covered by MISR (446, 558, 672, and 867 nm) and CATS (1024 nm), it is extremely challenging to convert the DAODs measured by these two instruments to the same wavelength for a quantitative comparison. Indeed, such conversion requires the currently lacking knowledge about the spectral dependence of dust, which further depends on dust properties that vary by dust source. Therefore, we compare spatial distributions of DAOD from



CATS and MISR in a semi-quantitative manner and report the spatial rank correlation between seasonal mean DAOD from
these two instruments.

Over the previously less explored Southern Hemisphere, the spatial distribution seasonal mean DAOD from CATS is further compared with CALIOP. Because dust is more sensitive in the visible bands, here we analyze dust extinction profiles at 532 nm from the CALIOP Version 4.10 Level 2 aerosol data products (Kim et al., 2018). CALIOP DAOD are calculated as the vertical integral of extinction over "dust" pixels. Given the CALIPSO overpass at local time 1:30 and 13:30, CATS
data averaged over 0:00-3:00 and 12:00-15:00 is compared with nighttime and daytime CALIOP DAOD, respectively. Daytime and nighttime dust extinction profiles from CATS and CALIOP is also compared over the Bodélé Depression — the global leading dust source.

**2.4 ANOVA-based significance test of dust diurnal variability**

In the current study, the dust diurnal variability at each location and its statistical significance is estimated and tested
under an analysis of variance (ANOVA) framework (Fisher, 1992). To account for potential inconsistency in CATS data quality during the local daytime and nighttime periods, as demonstrated in section 3.1, the analysis is performed for day and night separately. Since we do not assess the full diurnal cycle, daytime and nighttime diurnal variation is defined here as the variation of mean DAOD within daytime period and nighttime period, respectively. At each pixel, the $k^{th}$ DAOD observation in local time window i and season j ($D_{ijk}$), is approximated as a sum of the global, annual mean DAOD (D), an annual mean
diurnal term ($d_i$), a seasonal term ($s_j$), a diurnal term varying by season ($ds_{ij}$), and an error term that reflects other factors ($\varepsilon_{ijk}$), namely:

$$D_{ijk} = D + d_i + s_j + ds_{ij} + \varepsilon_{ijk} \quad (1)$$

In order to achieve sufficient sample size, CATS DAOD is aggregated into each 3-hour local time window (0-3, 3-6, etc), in each season [December-February (DJF), March-May (MAM), June-August (JJA), and September-November
(SON)], in each 2° latitude x 2° longitude pixel, so that within each combination of pixel, time window, and season, there are at least 50 DAOD observations during 2015-2017.

The statistical significance of the diurnal variability, namely variability in $d_i's$, is determined through an F-test, following the classical two-way ANOVA variance partitioning approach. An F-statistic is constructed as

F = Variance of $d_i$'s / Variance of $\varepsilon_{ijk}$'s.  (2)

Under the null hypothesis that there is no diurnal variability in DAOD at a specific pixel, F follows an F-distribution with degrees of freedom (df₁, df₂) determined by the number of diurnal time windows (4 in the current study), number of seasons (4), and total number of observations in each pixel (assuming equal to n), with

$$df_1 = 4-1 \quad (3)$$

$$df_2 = n - 4 - 4 - 4x4 \quad (4)$$





Based on the value of the F-statistic and its degrees of freedom, a p-value can be determined, thereby determining the statistical significance of the diurnal variability. If the p-value is smaller than the pre-determined threshold (0.05 in the current study), it is very likely that the alternative hypothesis that there is diurnal variability is true. In the results and discussion sections, we mainly focus on regions with significant diurnal variability (p-value < 0.05). Note that in order to apply the F-test, the assessed variable is required to follow Gaussian distribution. Therefore, a log-transformation is

performed on the observed DAOD, and the application of the ANOVA framework is based on the logarithm of DAOD.

## 3 Results

### 3.1 Comparison of CATS and AERONET AOD

As an initial assessment of the potential quality inconsistency between daytime and nighttime CATS data, total AOD from CATS is evaluated against AERONET, and the agreement between CATS and AERONET AOD, in terms of root-

mean-square-error (RMSE), percentage of error within error range [%EE, namely ± (0.03+10% of AERONET AOD)], correlation (R), and mean bias, is compared among daytime and nighttime collocated observations at each AERONET site. According to the comparison (Figure 1), there is significant difference between daytime and nighttime CATS AOD quality. Nighttime CATS AOD observations exhibit apparently higher correlation (0.62), lower RMSE (0.11), and higher %EE (65) with respect to collocated AERONET AODs, than those of daytime observations (0.44, 0.18, and 57, respectively). These

differences are statistically significant according to the bootstrap test at a significance level of 0.05. Although the mean biases are similar between the daytime and nighttime samples, the significantly different correlation, RMSE, and %EE prevents a direct comparison between daytime and nighttime AOD or DAOD values. The currently identified difference in the quality of CATS daytime and nighttime retrievals was noted by previous studies (e.g. Yorks et al. 2016; Pauly et al. 2019). Indeed, Pauly et al. (2019) pointed out the high daytime lidar calibration uncertainty at 1064 nm (16-18%) with a

corresponding uncertainty of ∼21% in daytime total attenuated backscatter, which is significantly larger than the uncertainty in the nighttime total attenuated backscatter at the same wavelength (∼7%). Furthermore, both daytime and nighttime CATS AOD retrievals appear to underestimate the ground truth from AERONET at high AOD (Figure 1c,f), indicating degraded CATS data quality in the presence of high aerosol loading. When the mean AOD exceeds 0.3, the negative mean bias between CATS and AERONET is about -0.15, or 50% of mean AOD with both daytime and nighttime retrievals.

### 3.2 Comparison of DAOD from CATS with MISR and CALIOP

According to the comparison with MISR, CATS generally well captures the spatial variations in DAOD, with highest agreement during the boreal winter (Figure 2). Consistent features captured by both MISR and CATS DAOD fields include (1) annually high dust loading over the tropical eastern Atlantic ocean and the seasonally-varying meridional distribution of the maximum dust loading driven by dust transport from North Africa (Yu et al., 2020a), (2) seasonally enhanced dust



activity over the Middle East, Arabian Peninsula, and Arabian Sea in MAM and JJA associated with active frontal passage (Yu et al., 2015b) and Shamal events (Yu et al., 2016), respectively, and (3) elevated level of dustiness over the northern Pacific ocean in boreal spring due to enhanced dust emission and transport from the Taklamakan and Gobi deserts (Yu et al., 2019). The most pronounced inconsistency between MISR and CATS DAOD fields is the apparent underestimation of dust loading over land, especially over regions with high dust loading, such as North Africa, thereby leading to the moderate but

statistically significant ($p<0.01$ according to the Student's t-test) overall spatial rank correlation between the two instruments. This apparent underestimation of high dust loadings by CATS is consistent with the underestimation of high aerosol loadings, as discussed in section 3.1.

Over the Southern Hemisphere, both CATS and MISR exhibit significant dust loading over the tropical southeastern Atlantic Ocean during the southern African active fire season in boreal summer and autumn (Yu et al., 2020b). These

observed dust plumes likely originate from dust emission after wildfires in southern Africa (Mahowald et al., 2005; Wagenbrenner, 2017; Wagenbrenner et al., 2013). This seasonal dust loading is also present in the CALIOP daytime observations over nearby land (Supplemental Figure 3). However, potential misclassification between dust, smoke, and cloud could complicate this apparent presence of dust over this region (Graham et al., 2003). Another region of potential misclassification between dust and cloud is the Southern Ocean, where there is no obvious dust source nearby but all CATS,

MISR, and CALIOP indicate significant dust loading all year round (Figure 3, Supplemental Figure 3 and Supplemental Figure 4). The dust loading over the Southern Ocean is apparently even higher at night (Supplemental Figure 4), according to CALIOP, potentially corresponding to the nighttime enhancement in cloud cover (Noel et al., 2018).

**3.3 Global diurnal variability in DAOD**

In aware of the inconsistent data quality of daytime and nighttime CATS retrievals, the variations of DAOD among

different time windows are assessed separately for daytime and nighttime periods. As demonstrated in the global maps of CATS DAOD in each 3-hour local time window (Figure 3), the diurnal variability in dust loading is typically more pronounced over land than over ocean, likely due to the fact that dust over ocean is primarily transported from remote dust sources over land. Moreover, the variations in DAOD is more pronounced during the daytime than nighttime. Over the global terrestrial area during the daytime period, 32% of the 2˚x2˚ latitude/longitude pixels exhibit statistically significant

diurnal variability in DAOD ($p<0.05$), according to the ANOVA-based F-test, compared with 4% of the oceanic pixels. In sections 3.3.1-3.3.4, the daytime and nighttime diurnal cycle of DAOD over North Africa and Middle East, Asia, North America, and the Southern Hemisphere, as well as the underlying meteorological processes are discussed in detail. The results are presented as maps of timing and magnitude of diurnally maximum and minimum DAOD during daytime and nighttime (Figures 4-9) and diurnal cycle of mean DAOD and vertical profiles of dust extinction and dust fraction over five

key dust source regions during each season (Figures 10-13).


### 3.3.1 North Africa and Middle East

Over North Africa and the Middle East, 57% of the terrestrial area exhibit significant diurnal variability in dust loading during the daytime period, mostly corresponding to previously identified dust sources including the Bodélé Depression, West African El Djouf, Iraqi Desert, and Rub-al Khali Desert (Ginoux et al., 2001, 2010, 2012; Prospero et al., 2002; Yu et al., 2013, 2018) (Figures 4a,b, 10-13). Over the Bodélé Depression, the world's leading dust source region, the annual average DAOD varies diurnally from 0.12 to 0.24 during the daytime, and 0.11 to 0.19 during the nighttime. The daytime maximum dust loading occurs shortly after sunrise during 9 am – noon local time in all seasons except boreal autumn (Figure 4a, panel l of Figures 10-13), associated with a peak in wind speed (panel l of Figures 10-13) as a result of the breakdown of nocturnal low-level jet (Fiedler et al., 2013). The identified morning peak in DAOD over the Bodélé Depression is consistent with previous analyses based on geostationary satellite instrument SEVIRI (Chaboureau et al., 2007; Schepanski et al., 2009) and visibility observations (N'Tchayi Mbourou et al., 1997). The Rub-al Khali Desert in the central Arabian Peninsula displays a similar diurnal cycle of DAOD with the Bodélé Depression, with a morning peak hypothetically also associated with the break of nocturnal low-level jet. Over El Djouf, the second largest dust source in North Africa (Yu et al., 2018), the daytime DAOD exhibits a morning peak over the western sub-region, likely associated with the break of nocturnal low-level jet (Fiedler et al., 2013; Schepanski et al., 2009), and an afternoon peak over the eastern sub-region, likely associated with the enhanced deep convection due to surface heating (Heinold et al., 2013). The daytime variability in DAOD over the Iraqi Desert features a peak around local noon, most pronounced in boreal summer with enhanced wind speed by active Shamal (Figure 12m), consistent with previous station-based analysis of wind speed and dust storm frequency (Yu et al., 2016).

Nighttime diurnal variability in North Africa and Middle East occurs significantly over only 18% of the terrestrial area in North Africa and Middle East, most pronounced over the Grand Erg Occidental and Grand Erg Oriental in Algeria (Figure 4c, d). The nighttime DAOD reaches its maximum value shortly after local midnight over both deserts, and reaches the minimum value shortly after sunset over the Grand Erg Occidental and before midnight over the Grand Erg Oriental. The nighttime mean DAOD ranges, namely 0.07-0.13 (nighttime minimum to maximum) across the Grand Erg Occidental and 0.07-0.09 across the Grand Erg Oriental, are much smaller than those observed over the Bodélé Depression during the daytime period. The meteorological driver of the nighttime dust variability in these regions are likely associated with the diurnal evolution of local pressure systems and the resulting strength of the Harmattan wind (Schepanski et al., 2017).

The vertical profile of dust extinction over the key dust sources in North African and Middle East largely reflects the diurnal and seasonal variations of planetary boundary layer (PBL), as the majority of dust particles are confined to PBL over these dust source regions (panels a-c of Figures 10-13). For example, over the Bodélé Depression, the highest daytime dust extinction in boreal spring occurs during 9 am - noon local time within 1 km above ground (Figure 11g). While in boreal summer, the high dust extinction above 0.5 extends from surface to 4 km above ground over the Bodélé Depression (Figure 12g), contributing to the trans-Atlantic dust transport in the Saharan Air Layer (Gasteiger et al., 2017). The percentage of





observations when dust is identified as the dominant aerosol feature (hereafter referred to as dust fraction), however, demonstrates the deteriorated data quality of CATS feature identification during the daytime. Typically less than half of the observed daytime aerosol is identified as dust, regardless of season and height, over the key dust sources in the El Djouf, Bodélé Depression, and Iraqi Desert, compared with above 70% of the nighttime observations featured as dust. This day-night inconsistency in dust fraction indicates higher chance of misclassifying dust to other aerosol types or clouds during the daytime. Given the higher chance of misclassification during the daytime, the daytime DAOD by CATS is hypothesized to underestimate the actual dust loading to a larger extent than the nighttime over the major dust source regions.

Furthermore, the comparison between dust extinction profiles from CATS and CALIOP at the Bodélé Depression (Supplemental Figure 5) confirms the existence of "false clear air" near surface in CATS data as identified by Lee et al. (2019). The dust extinction from CATS is substantially lower than CALIOP below 1 km AGL during daytime and below 500 m AGL during nighttime. The daytime underestimation of CATS dust extinction could also be attributed to the aforementioned higher chance of the misclassification.

### 3.3.2 Asia

In contrast to the North African and Middle Eastern dust sources, the majority of the terrestrial area in Asia exhibit insignificant dust diurnal variability during either daytime or nighttime periods (Figure 5). Two of the major dust sources in Asia, namely the Thar Desert in India and Taklamakan Desert in China, both show spatially inhomogeneous and statistically insignificant dust diurnal variability. According to simulations from a regional climate model, the dominant source of dust over the Thar desert varies by season (Banerjee et al., 2019). While local dust sources provide the majority of dust during the boreal summer monsoon season, the remote sources in the Middle East, Arabian Peninsula, and West Asia contribute comparable amount of dust than the local sources to the northern India during the rest of the year. Since the diurnal cycle of transported dust is less deterministic than locally emitted dust, the overall annual mean diurnal cycle over the Thar Desert is less robust than that over the dust source regions in North Africa. In boreal summer, however, the daytime DAOD over the Thar Desert exhibits a significant morning peak at about 0.35 (Figure 12n), consistent with the implications from the modelling study (Banerjee et al., 2019). Over the Taklamakan Desert in China, although the mean range between the diurnally maximum and minimum DAOD reaches about 20% of the long-term average, quantitatively consistent with a previous AERONET-based assessment at a nearby site (Wang et al., 2004), the diurnal cycle in DAOD is mostly insignificant because the regional dust emission is mainly driven by the frontal passage, which does not have a clear diurnal cycle (Luo et al., 2004).

Similar with dust sources in North Africa and Middle East, dust fraction over the Thar and Taklamakan deserts is typically lower during the daytime than during the nighttime period, regardless of height and season (panels d and e in Figures 9-12). The contrast between daytime and nighttime dust fraction over the Asian dust source regions is smaller than that over the North African and Middle Eastern dust source regions. On the other hand, the vertical profile of dust extinction suggests seasonally varying relative contribution of local versus remote dust sources. For example, over the Thar Desert, the





summertime high dust extinction above 0.5 extends from surface to 6 km above ground (Figure 11i). While in boreal spring and autumn, the high dust extinction presents only at several kilometres above ground (panel i in Figure 10 and 12). This seasonal contrast in dust extinction profile is consistent with the modelling-based finding about the dominance of local dust source in the Thar Desert in boreal summer (Banerjee et al., 2019).

### 3.3.3 North America

Over North America, about 21% and 9% of the terrestrial area exhibits statistically significant dust diurnal variability during the daytime and nighttime periods, respectively, although both the mean and diurnal range of DAOD are typically less than 0.03, much smaller than those over North Africa, the Middle East, and Asia (Figure 6). During the daytime, an afternoon peak in DAOD at about 0.08 spreads across the southern and western sub-region of the continent, yet the majority of the continent exhibits insignificant diurnal variability during either the daytime or nighttime period. The identified afternoon peak in DAOD is consistent with previous visibility-based and meteorology-based observational analysis, which identified deep convection as the dominant process driving the dust emission across North America (Stout, 2015).

### 3.3.4 Southern Hemisphere

The Southern Hemisphere exhibits generally weak dust loading, especially in terms of diurnally minimum DAOD (Figures 7-9), with significant dust diurnal variability over the southwestern Amazon Rainforest and southwestern Congo Rainforest during daytime, as well as central Australia during nighttime. Since there have been limited exploration of the diurnal dust cycle over these regions, hypotheses are provided here regarding the underlying driving processes of these diurnal dust variations.

Over the Amazon Rainforest, it has been believed that remote dust sources in North Africa provide key nutrients to fertilize the Amazon Rainforest through trans-Atlantic dust transport (Kaufman et al., 2005; Koren et al., 2006; Yu et al., 2015a). However, since the trans-Atlantic dust transport typically takes several days to weeks, it is unlikely that the amount of transported dust displays a clear diurnal cycle, as indicated by CATS DAOD (Figure 7). On the other hand, field observations and model simulations have identified a potential contribution from local, post-fire dust emission over vegetated area (Mahowald et al., 2005; Wagenbrenner, 2017; Wagenbrenner et al., 2013). Similarly, significant daytime diurnal cycle of DAOD is hypothesized to be associated with post-fire dust emission over the southern boundary of the Congo Rainforest (Figure 8). In particular, the observed burned fraction has substantially increased across the Amazonia and tropical southern Africa during the recent years (Andela et al., 2017), which might be responsible for the small-magnitude, yet statistically significant diurnal variability over these regions.

Over the central Australia, DAOD exhibits diurnal variability only during the nighttime period, with a general peak in the early evening (Figure 9). Dust emission in the central Australia is believed to be driven by frontal passage, similar with the Taklamakan Desert in China (Knippertz and Stuut, 2014). However, over central Australia, low-level winds associated



with cold fronts intensify with the deformation and convergence in the early evening, driven by the subsidence of the mixing layer (Thomsen et al., 2008), thereby leading to an enhanced dust emission at that time of the day.

## 4 Conclusions and Discussion

Based on the profiles of aerosol typing and extinction observed by Cloud-Aerosol Transport System (CATS) on the International Space Station (ISS), the diurnal cycle of dust loading over the global tropics, subtropics, and mid-latitudes is quantified, with the statistical significance of dust diurnal variability determined through the Analysis of Variance (ANOVA)-based F-test. Based on the comparison between Aerosol Optical Depth (AOD) derived from CATS and Aerosol Robotic Network (AERONET), daytime and nighttime CATS retrievals exhibit significant difference in the data quality, thereby supporting the analysis of separate diurnal variations during daytime and nighttime periods using CATS data. The spatial variations in dust AOD (DAOD) are reasonably captured by CATS, according to the comparison with the nonspherical, dust AOD from the Multi-angle Imaging SpectroRadiometer (MISR). The analytical framework yields statistically robust findings about the dust diurnal cycle that is generally more pronounced during daytime periods and over terrestrial areas. The currently identified diurnal cycle in DAOD confirms previous geostationary-based, and ground-based observational conclusions in key dust source regions, including (1) the Bodélé Depression, Rub-al Khali Desert, and western El Djouf, which exhibit a morning peak in DAOD driven by the break of nocturnal low-level jet, and (2) the eastern El Djouf, and the southern and western North America, which exhibit an afternoon peak in DAOD, driven by an enhanced deep convection. Insignificant diurnal cycle in DAOD are found over these dust source regions: (1) the Iraqi Desert, where the noon peak in DAOD associated with the enhanced pressure gradient driven by spatially differential heating is only robust in boreal summer, (2) the Thar Desert, where the dominant source of dust varies by season and the diurnal variation in DAOD is only significant in boreal summer when the local source dominates over remote resources, and (3) the Taklamakan Desert, where dust emission is primarily driven by frontal passage which does not exhibit a clear seasonal cycle. Over the Southern Hemisphere, it is hypothesized that post-fire dust emission is likely responsible for the maximum dust loading over the rainforests in Amazon and tropical southwestern Africa.

By comparing the CATS AOD and DAOD with that from AERONET and MISR, respectively, underestimation by CATS in the presence of high aerosol or dust loading are identified here. Previous studies based on Cloud-Aerosol Lidar with Orthogonal Polarization (CALIOP) observations noted that the laser backscatter signal becomes totally attenuated at particulate column optical depths of about 3, so that there are occasions where lidars, such as CATS and CALIOP, cannot measure the full extent of the vertical column in the thick dust layer (Vaughan et al., 2009). Furthermore, the CATS feature detection algorithm creates a gap between the surface and near-surface aerosol base altitude, causing false regions of "clear-air" between the surface and near-surface aerosol layers despite the possible presence of aerosols in this altitude region. CATS does not use an aerosol base extension algorithm, like CALIOP, that detects scenarios when aerosols are present in the bins just above the surface and extends the near-surface aerosol layer base down to the surface (Tackett et al., 2018;





385 Yorks et al., 2019). The complete attenuation and feature detection problems likely lead to underestimation of the near surface dust extinction by CATS over dust source regions, such as the El Djouf, Bodélé Depression, and Middle East (Figures 10-13), underestimation of DAOD as compared with MISR (Figure 2), as well as underestimation of total AOD as compared with AERONET (Figure 1). Furthermore, higher chance of misclassifying dust to cloud or other aerosols also contributes to the underestimation of DAOD in the current analytical framework, as discussed in sections 3.3.1-3.3.2.

390 The diurnal cycle of dust emission and dust loading over the Southern Hemisphere requires further investigation. The dust sources in the Southern Hemisphere provide key nutrients to the oceans in the Southern Hemisphere, and thereby playing an important role in the global biogeochemical cycle (Mahowald et al., 2011). However, the Southern Hemispheric dust sources received much less research efforts than the Northern Hemispheric dust sources in the past. Beyond the hypotheses regarding the dust diurnal variability and the underlying driving processes raised in Section 3.3.4, influence from 395 potential uncertainty in the CATS aerosol loading and typing retrievals, such as misclassification between dust, smoke, and biological particles (Graham et al., 2003) and presence of aerosols under optically thick cloud layers, needs to be investigated from other sources of observation to validate the currently assessed diurnal cycle in dust loading over the Southern Hemisphere. Furthermore, the current study examines the dust and dust mixture types reported by the CATS classification algorithm. Such algorithms rely partly on depolarization ratio, but the results are likely contaminated by other 400 aerosol types that exhibit overlapping depolarization ratio ranges with dust (Burton et al., 2015; Freudenthaler et al., 2009; Haarig et al., 2017). Future studies are encouraged to take the advantage of methodologies that have been developed to address the decoupling of dust and non-dust components (Amiridis et al., 2013; Mamouri and Ansmann, 2014; Tesche et al., 2009).

Beyond uncertainties in the CATS retrieval algorithm, sampling is a major source of uncertainty in the current study. As 405 demonstrated in the number of overpasses per region per three-hour window over the five key dust source regions (Figures 10-13), the number of seasonal overpasses varies from 4 to 27 by season, hour, and region during the 2.5-year CATS period. Over North Africa, CATS samples the nighttime period more frequently than daytime, leading to even higher uncertainty during the daytime given its degraded daytime data quality. The insufficient sampling causes potential underrepresentation of the less frequently active dust source regions, such as those located in the Southern Hemisphere. According to a 410 comparison between the 3-year and 11-year CALIOP DAOD maps over the Southern Hemisphere (Supplemental Figure 3 and Supplemental Figure 4), the longer record results in a more stable and smooth climatology of dust loading. In aware of this sampling uncertainty in the current study, future studies on global diurnal variability will substantially benefit from potentially longer CATS-like observations.



**Author Contribution**

YY led the study with inputs from all coauthors. OVK, MJG, JEY, and JRC provided guidance on the satellite data processing. JM processed the CALIOP data. HL and MC contributed to the statistical analysis. GSO helped with the results interpretation. YY prepared the manuscript with contributions from all coauthors.

**Acknowledgements**

CATS L2 aerosol profile data and MISR L3 aerosol data were obtained from the NASA Langley Research Center Atmospheric Science Data Center. AERONET V3 AOD data was obtained from the AERONET website (https://aeronet.gsfc.nasa.gov/). This work was partially performed at the Jet Propulsion Laboratory, California Institute of Technology, under a contract with the National Aeronautics and Space Administration. The authors thank the MISR team for providing facilities and useful discussions.

**Data availability**

The research data used are available at the sources specified in the acknowledgement section.

**Competing interests**

The authors declare that they have no conflict of interest.

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

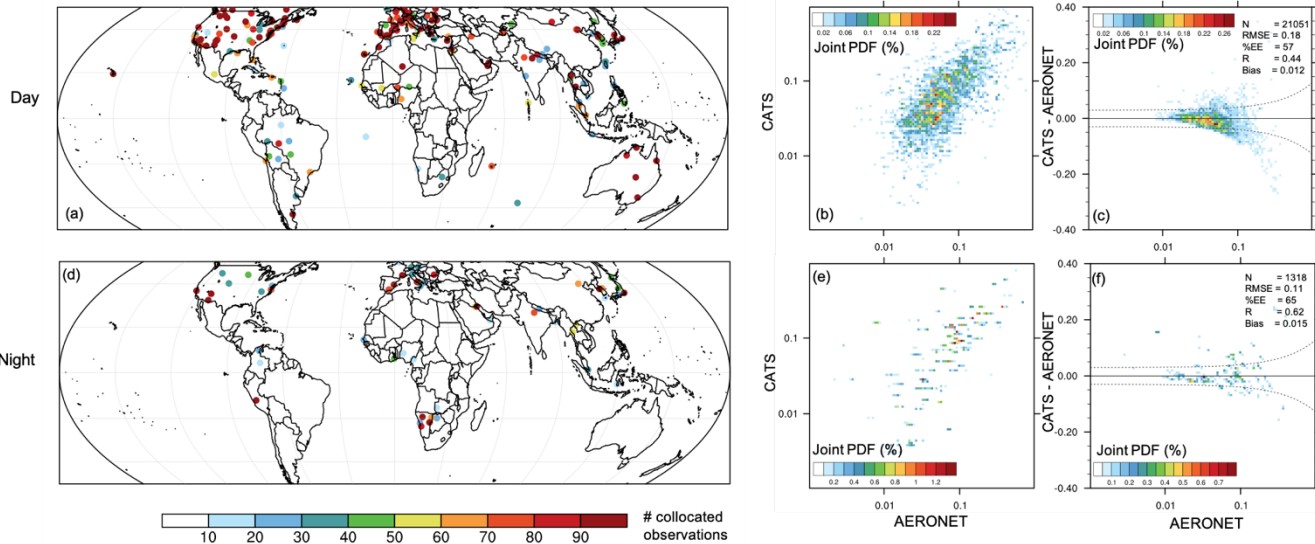

**Figure 1 Comparison of total AOD from CATS (1064 nm) and AERONET (1020 nm) during the local (a-c) day and (d-f) night. (a, d) number of collocated observations between CATS and AERONET. (b, e) Joint probability density (PDF, %) of collocated AOD from CATS and AERONET. (c, f) AOD difference between CATS and AERONET as a function of AERONET AOD. The dashed lines in (c) and (f) represent the uncertainty range of AERONET AOD. The overall collocation sample size (N), root-mean-**

**squared-error (RMSE), percentage of error within in AERONET uncertainty range (%EE), correlation coefficient (R), and mean bias for daytime and nighttime observation are indicated in (c) and (f), respectively.**



**Figure 2 Comparison of seasonal average, standardized DAOD from (a-d) CATS (1064 nm) and (e-h) MISR (558 nm) in (a, e) December-February (DJF), (b, f) March-May (MAM), (c, g) June-August (JJA), and (d, h) September-November (SON) during 2015-2017. For a direct comparison, DAOD from each instrument in each season is standardized, namely divided by the 95th percentile of all DAOD observations between 51°N-51°S from that instrument in that season and multiplied by 100. The spatial rank correlation between the seasonal DAOD maps from CATS and MISR are indicated in the corresponding CATS panel.**


**(a) 00-03**   **(b) 03-06**

**(c) 06-09**   **(d) 09-12**

**(e) 12-15**   **(f) 15-18**

**(g) 18-21**   **(h) 21-24**

0.03   0.06   0.09   0.12   0.15   0.18   0.21   0.24

**Figure 3 Annual-average DAOD from CATS during 2015-2017 in each 3-hour local time window: (a) 00-03, (b) 03-06, (c) 06-09, (d) 09-12, (e) 12-15, (f) 15-18, (g) 18-21, (g) 21-24.**



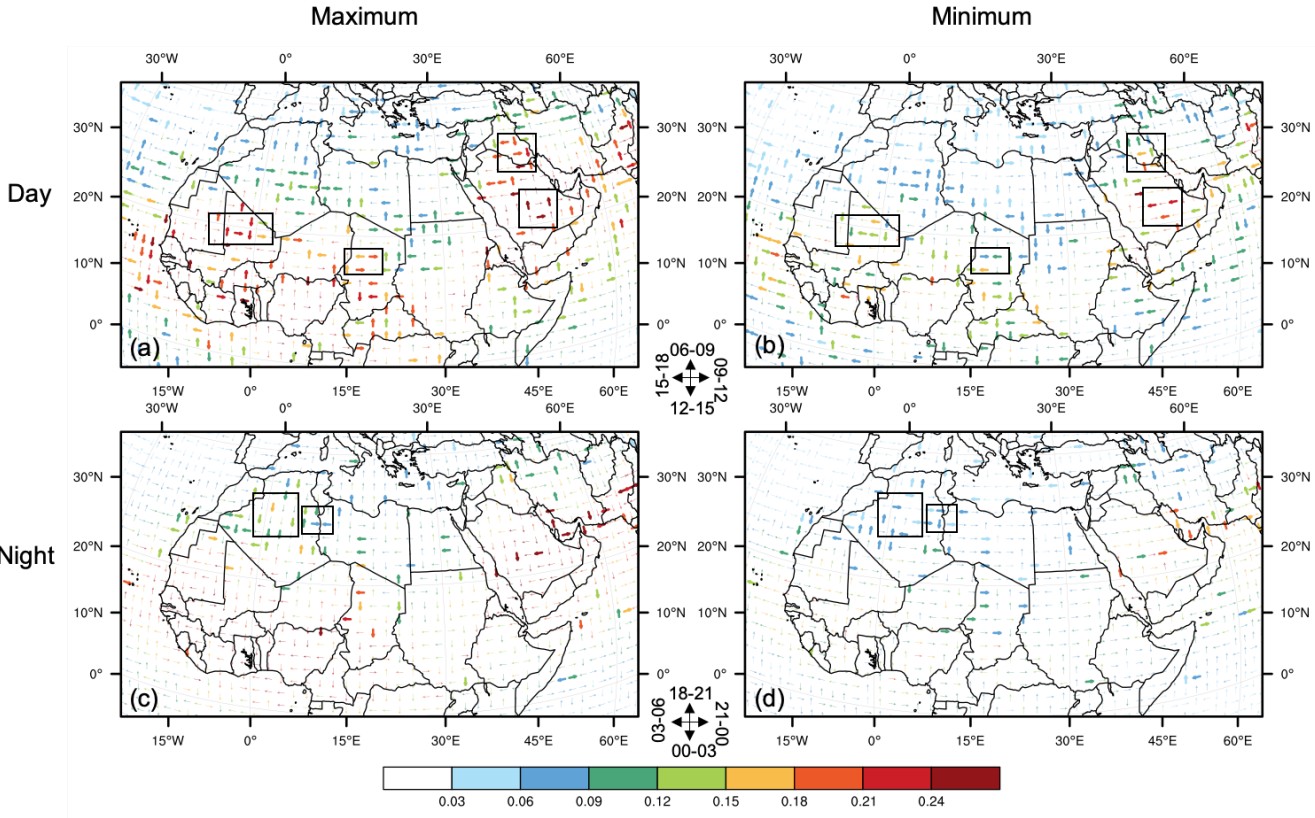

**Figure 4 Diurnally (a, c) maximum and (b, d) minimum DAOD across North Africa and Middle East during (a, b) daytime and (c, d) ngihttime periods. Direction of the vectors indicates the local time when mean DAOD reaches the (a, c) maximum and (b, d) minimum value during the (a, b) day and (c, d) night. Color of the vectors indicates the (a) maximum and (b) minimum DAOD value. Thick vectors indicate statistically significant diurnal variability ($p<0.05$) according to the ANOVA-based F-test. In (a) and (b), the boxes show the location of the West African El Djouf, Bodélé Depression, Iraqi Desert, and Rub-al Khali Desert from west to east. In (c) and (d), the boxes indicate the Grand Erg Occidental and Grand Erg Oriental from west to east.**







**Figure 5 Diurnally (a, c) maximum and (b, d) minimum DAOD across Asia during (a, b) daytime and (c, d) ngihttime periods.**
**Figure elements are the same as in Figure 4. The boxes in (a) and (b) show the location of Thar and Taklamakan deserts from west**
**to east.**





**Figure 6 Diurnally (a, c) maximum and (b, d) minimum DAOD across North America during the (a, b) daytime and (c, d) nighttime periods. Figure elements are the same as in Figure 4.**




**Figure 7 Diurnally (a, c) maximum and (b, d) minimum DAOD across South America during the (a, b) daytime and (c, d) nighttime periods. Figure elements are the same as in Figure 4.**




**Figure 8** Diurnally (a, c) maximum and (b, d) minimum DAOD across South Africa during the (a, b) daytime and (c, d) nighttime periods. Figure elements are the same as in Figure 4.





**Figure 9 Diurnally (a, c) maximum and (b, d) minimum DAOD across Australia during the (a, b) daytime and (c, d) nighttime periods. Figure elements are the same as in Figure 4.**



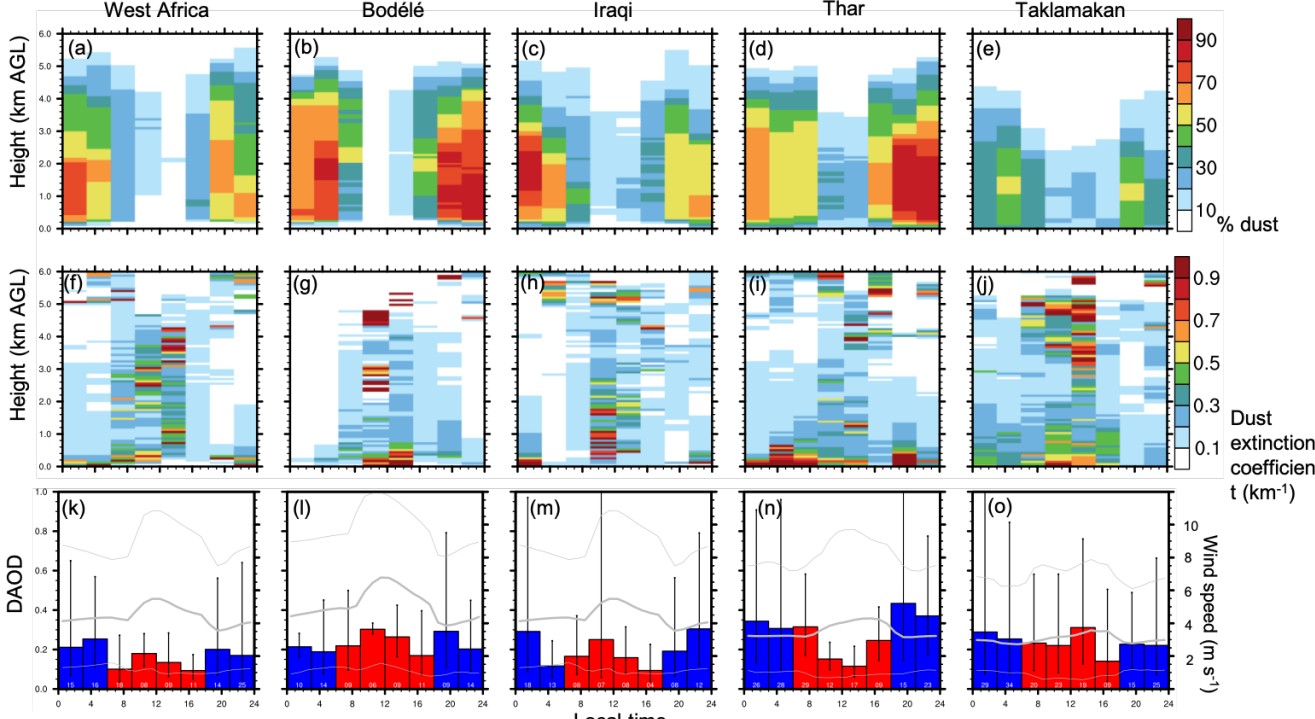

**Figure 10 Diurnal cycle of dust characteristics observed by CATS over five key dust source regions during boreal winter (December-February, DJF). (a-e) Regional average percentage of observations when dust is identified as the dominant aerosol feature by height. (f-j) Mean dust extinction by height. (k-o) Regional mean DAOD (average across multiple overpasses: bars, with red and blue bars representing daytime and nighttime observations, respectively; minimum and maximum across multiple overpasses: vertical lines) referring to the left Y-axis, and regional mean 10-m wind speed (m s⁻¹, grey lines), referring to the right Y-axis. The hourly wind speed data is obtained from the WATCH Forcing Data for ERA5** (Beck et al., 2016; Cucchi et al., 2020)**. The thick and thin lines represent the average, minimum, and maximum wind speed during the DJF of 2015-2017. Number of CATS overpasses per region per time window is indicated in (k-o). The location of each dust source region is indicated in Figures 4 and 5.**






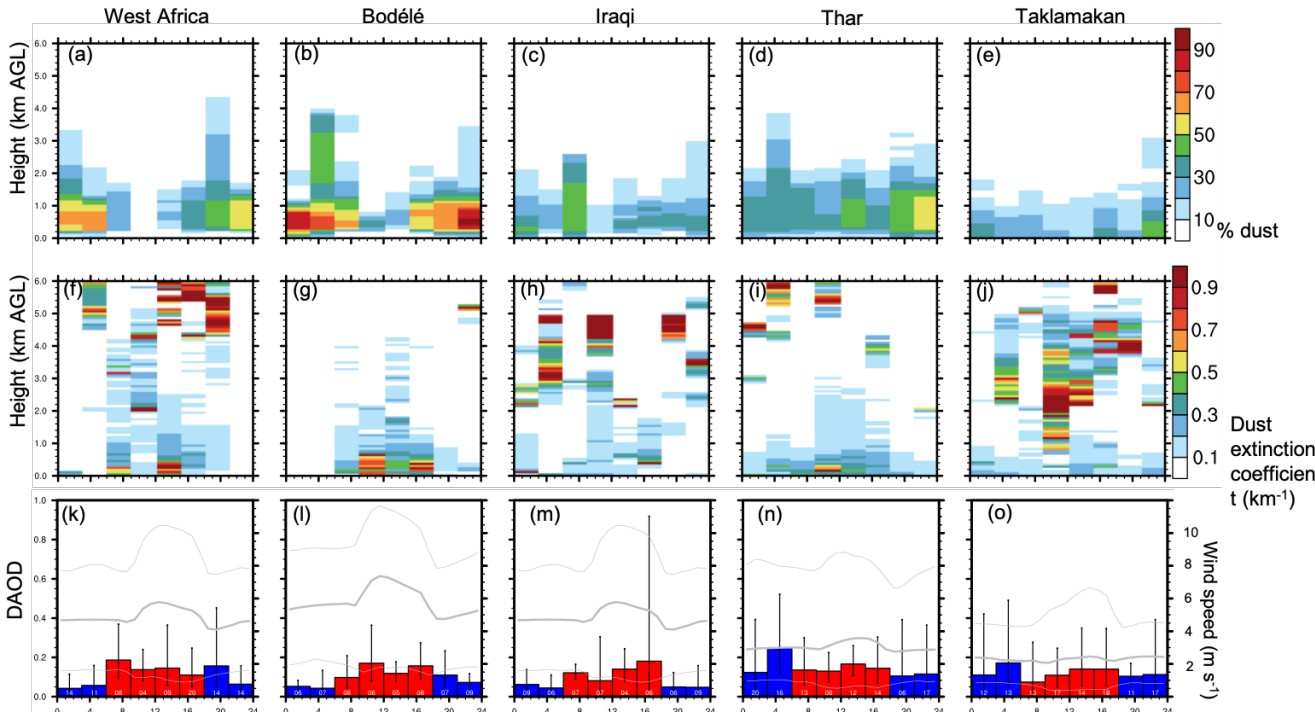

**Figure 11 Diurnal cycle of dust characteristics observed by CATS over five key dust source regions during boreal spring (March-May, MAM). Figure elements are the same as in Figure 10.**






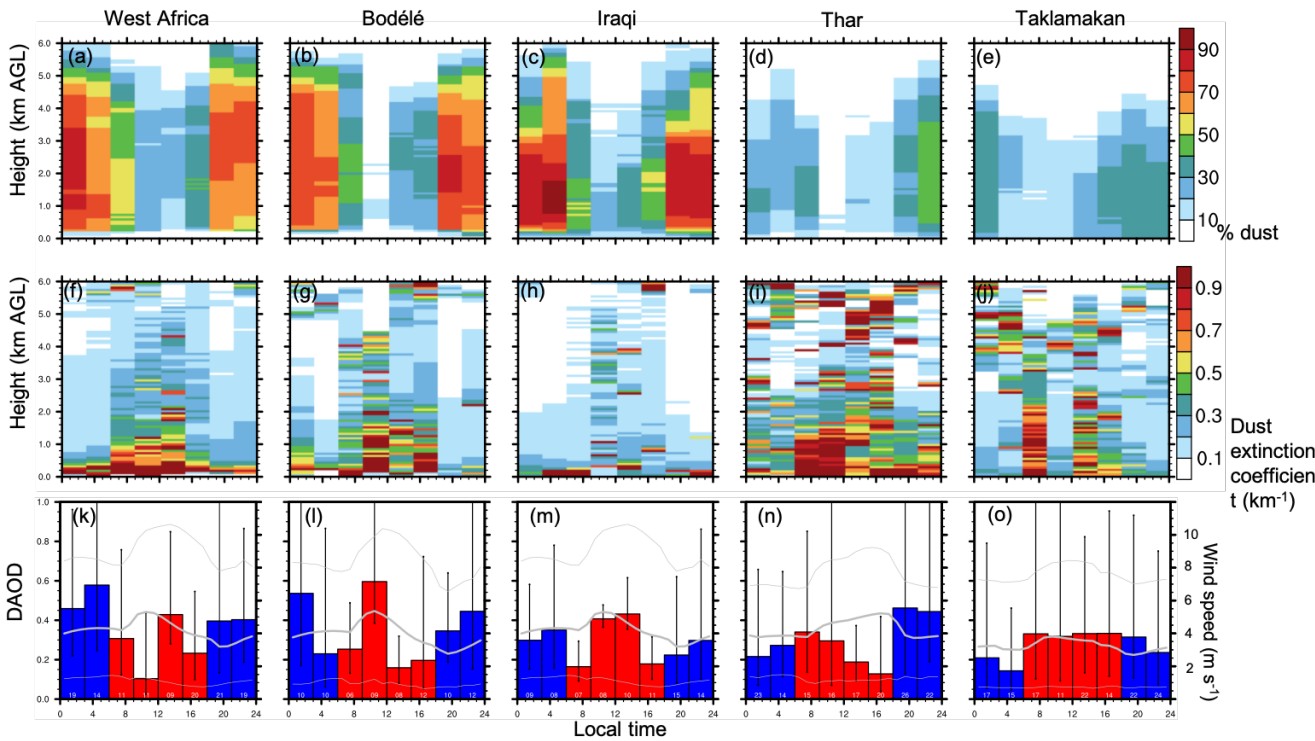

**Figure 12 Diurnal cycle of dust characteristics observed by CATS over five key dust source regions during boreal summer (June-August, JJA). Figure elements are the same as in Figure 10.**






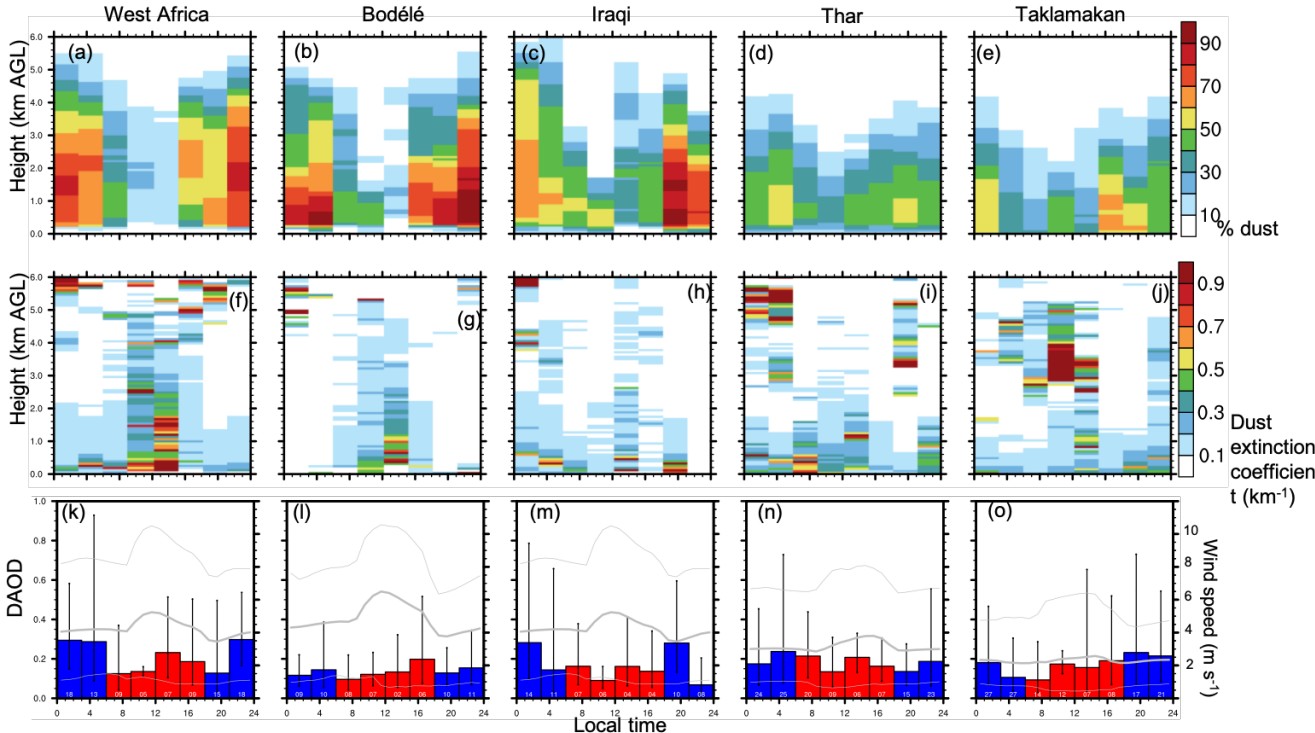

**Figure 13 Diurnal cycle of dust characteristics observed by CATS over five key dust source regions during boreal autumn (September-November, SON). Figure elements are the same as in Figure 10.**
