# Peer review of "A Global Analysis of Diurnal Variability in Dust and Dust Mixture Using CATS Observations"

_Atmospheric Chemistry and Physics, 2020_

## Referee Comment (RC1) · Anonymous Referee #2 · 29 Oct 2020

This paper deals with the diurnal variations of dust using measurements from the CATS lidar that operated on International Space Station between 2015 and 2017. The manuscript is a revised version of an earlier submission to ACPD. The primary problem with that submission was that the authors did not address the substantial difference in the CATS lidar calibration from daytime to nighttime that is known to exist. In the current version the authors have attempted to address this issue by analyzing the daytime and nighttime variations in local time separately. This is not quite a satisfactory way of dealing with the problem. Thus no attempt has been made to disentangle the daytime variations from the large uncertainties (∼21%) in total attenuated backscatter coming from the large daytime calibration uncertainties. This may complicate interpretations of the variability with low dust loadings, for instance over North America as well as in

the southern hemisphere. Also they continue to use the quality unassured and sparse AERONET nighttime data to assess CATS day/night data quality. In any case the current version has improved particularly in terms of including data from CALIOP as well as providing vertical and seasonal information of the variations and including wind data to interpret the dust loading. They have also revised the manuscript in the light of other comments from the referees on the previous version. The paper is within the scope of Atmospheric Chemistry and Physics and should be useful to the community interested in dust variability using satellite measurements. I have a few minor comments on the current version.

1. There are some significant differences between the CATS plots of seasonal DAOD in Figure 2 and supplemental Figure 3. For example, note the high DAOD values of ∼0.2 over the biomass burning areas in southern Africa in JJA in the supplemental Figure 3, which are not there in the corresponding plots in Figure 2. If this is due to the standardization applied to Figure 2 then it should be made clear in the text in section 3.2. I do not see particularly high values of DAOD in the corresponding CALIOP 532 nm plots either for 2015-2017 or 2006-2017. The authors have used dust as well as mixture of dust with smoke or marine aerosols, i.e polluted dust and dusty marine in CALIOP terminology. On the other hand they have used only "dust" from CALIOP data—could this be making any difference in these plots?

2. Does Figure 2 include data from both day and night?

3. Lines 119-120: this sentence needs to be rephrased, may be replacing "According to . . .modelling results" by "For example,.." and giving the link to the browse image rather than just giving the NASA center name : https://cats.gsfc.nasa.gov/data/segment_detail/330280/

4. There are some inconsistencies between the text (lines 321-323) and the plot numbers referred to.

5. Line 297 and supplemental Figure 5: please specify the CALIOP wavelength used

in this Figure —is it still 532 nm?

6. In Figures 10-13, the number of CATs overpasses within each time window is shown at the bottom of the bars but is barely visible. Please increase the font size or use bold.

---

## Referee Comment (RC2) · Anonymous Referee #1 · 5 Nov 2020

The paper "Global Analysis of Dust Diurnal Variability Using CATS Observations" by Yan Yu et al. investigates the diurnal cycle of dust loading across the global tropics, sub-tropics, and mid-latitudes by analyzing aerosol extinction and typing profiles observed by CATS lidar aboard the ISS. CATS was developed to address three main science objectives; with one of the goals to measure and characterize aerosols/clouds on a global scale and at various local times. The diurnal variability of aerosols consists a significant scientific question. The present study attempts to build on the aerosol diurnal study performed by Lee et al., 2018, focusing on dust aerosols. The idea of the study is of high scientific interest, falls within the scope of ACP, the manuscript is well-structured, the presentation clear, the language fluent.

The authors have addressed the daytime underestimations compared to nighttime ob-

servations through analyzing separately CATS-ISS daytime and nighttime observations, and removing comments unifying the daytime and nighttime variability. In addition, vertical mean extinction coefficient profiles have been included, a significant advantage of lidar systems. However, the manuscript would substantially improve though including additional information on the profiles of extinction coefficient variability in the figures.

One major comment that is still not addressed corresponds to the implementation of the aerosol extinction coefficient of "dust" and "dust mixture" in the analysis, both interpreted as dust and accounted as DAOD. Although there is the full dust component included, the dust component coexists in the analysis with other aerosol types, contaminating the pure-dust assumption. Thus, the study does not correspond to the diurnal variability of "dust" but to the variability of "dust mixture". In case the study is not adapted, in both study name and context when referring to "dust" or "DAOD", the study will be misleading to the research community, for the statement that it provides the diurnal variability of dust is very strong and non-onsistent with the content and approach. I strongly suggest the authors to revise the manuscript accordingly, prior proceeding with publication.

---

## Author Comment (AC1) · 13 Dec 2020

The comment was uploaded in the form of a supplement:
https://acp.copernicus.org/preprints/acp-2020-991/acp-2020-991-AC1-supplement.pdf

---

## Author Response (AR1)

Dear Dr. Tesche and two reviewers,

Thank you for your valuable comments and suggestions. Corresponding to your suggestions, we made the following major changes in the manuscript:
1) Emphasizing that the current study assesses the diurnal variability in dust and dust mixtures
2) Separately comparing CATS' and CALIOP's representation of pure dust and dust mixtures
3) Including variability of dust extinction profiles for key dust source regions in Figures 10-13
Responses to each of your comments follow below.

Yan on behalf of all authors

RC1

The paper "Global Analysis of Dust Diurnal Variability Using CATS Observations" by Yan Yu et al. investigates the diurnal cycle of dust loading across the global tropics, sub-tropics, and mid-latitudes by analyzing aerosol extinction and typing profiles observed by CATS lidar aboard the ISS. CATS was developed to address three main science objectives; with one of the goals to measure and characterize aerosols/clouds on a global scale and at various local times. The diurnal variability of aerosols consists a significant scientific question. The present study attempts to build on the aerosol diurnal study performed by Lee et al., 2018, focusing on dust aerosols. The idea of the study is of high scientific interest, falls within the scope of ACP, the manuscript is well-structured, the presentation clear, the language fluent.

Thank you for the positive feedback on our revised manuscript. We deeply appreciate your comments and suggestions during the last round of review.

The authors have addressed the daytime underestimations compared to nighttime observations through analyzing separately CATS-ISS daytime and nighttime observations, and removing comments unifying the daytime and nighttime variability. In addition, vertical mean extinction coefficient profiles have been included, a significant ad- vantage of lidar systems. However, the manuscript would substantially improve though including additional information on the profiles of extinction coefficient variability in the figures.

Thank you for recommending the additional information on the variability of the extinction profiles. The variability in dust and dust mixture extinction coefficient over key dust source regions are included in the revised Figures 10-13, and discussed on lines 421-425 of the revised manuscript, read as "*The limited sampling results in the large spread of observed dust/dust mixture extinction (panels f-j of Figures 10-13) and DAOD (panels k-o of Figures 10-13) over the key dust sources. The spread in both extinction profiles and DAOD appears larger in less dusty regions, namely the Thar and Taklamakan Deserts, than the dustier regions, namely the El Djouf and Bodélé Depression*".

One major comment that is still not addressed corresponds to the implementation of the aerosol extinction coefficient of "dust" and "dust mixture" in the analysis, both interpreted as dust and accounted as DAOD. Although there is the full dust component included, the dust component coexists in the analysis with other aerosol types, contaminating the pure-dust assumption. Thus, the study does not correspond to the diurnal variability of "dust" but to the variability of "dust mixture". In case the study is not adapted, in both study name and context when referring to "dust" or "DAOD", the study will be misleading to the research community, for the statement that it provides the diurnal variability of dust is very strong and non-consistent with the content and approach. I strongly suggest the authors to revise the manuscript accordingly, prior proceeding with publication.

Thank you for bringing up the uncertainty in our study regarding the dust versus dust mixture. We emphasize throughout the revised manuscript that this study assesses the diurnal variability in dust and dust mixture. This change is reflected in the revised title, "*A Global Analysis of Diurnal Variability in Dust and Dust Mixture Using CATS Observations*", and abstract, among many other sections of the revised manuscript.

The revised abstract reads as "*The current study investigates the diurnal cycle of dust and dust mixture loading across the global tropics, sub-tropics, and mid-latitudes by analyzing aerosol extinction and typing profiles observed by the Cloud-Aerosol Transport System (CATS) lidar aboard the International Space Station. According to the comparison with ground-based and other satellite observations, CATS aerosol and dust/dust mixture loading observations exhibit reasonable quality but significant day-night inconsistency. To account for this day-night inconsistency in CATS data quality, the diurnal variability in dust/dust mixture characteristics is currently examined separately for daytime and nighttime periods. Based on an analysis of variance analytical framework, pronounced diurnal variations in dust/dust mixture loading are generally uncovered during daytime periods and over terrestrial areas. The current study identifies statistically significant diurnal variability in dust/dust mixture loading over key dust sources, including the Bodélé Depression, the West African El Djouf, Rub-al Khali Desert, and western and southern North America, confirming the previous observation-based findings regarding the diurnal cycle of dust emission and underlying meteorological processes in these regions. Significant seasonal dust/dust mixture diurnal variability is identified over the Iraqi and Thar deserts. The identified significant diurnal cycles in dust mixture loading over the vegetated regions in Amazon and tropical southern Africa are hypothesized to be driven by enhanced dust emission due to wildfires*".

We now specifically point out that the current study assesses dust and dust mixture, on lines 133-135 of the revised manuscript, read as "*Note that over regions where dust is less dominant, the total concentration of dust and dust mixture aerosols does not always reflect the abundance of dust particles. Therefore, the currently assessed DAOD represents an upper limit of the dust loading sampled by CATS*".

In addition, we expand the comparison between CALIOP and CATS by separating pure dust (revised Supplemental Figures 3 and 5) from dust and dust mixture (revised Supplemental Figures 4 and 6). This comparison is outlined on lines 170-174 of the revised manuscript, read as "*To be consistent with CATS, CALIOP DAOD are calculated as the vertical integral of extinction over "dust", "polluted dust", and "dusty marine" pixels. Furthermore, the pure DAOD (pDAOD), namely the vertical integral of extinction over "dust" pixels, from both CATS and CALIOP is compared, thereby shedding light on the quantitative difference between pure dust and dust mixture loadings over the less dusty Southern Hemisphere*". The uncertainty from including both dust and dust mixture for the Southern Hemisphere is discussed on lines 242-246 of the revised manuscript, read as "*These observed dust plumes likely originate from dust emission after wildfires in southern Africa (Mahowald et al., 2005; Wagenbrenner, 2017; Wagenbrenner et al., 2013), and/or are likely due to higher amount of biomass burning aerosols mixed with dust. This seasonal dust/dust mixture loading is also present in the pDAOD in both CATS and CALIOP daytime observations over the tropical southeastern Atlantic and southern Africa (Supplemental Figure 3) and doubles when accounting for both dust and dust mixture according to both instruments (Supplemental Figure 4)*".

RC2

This paper deals with the diurnal variations of dust using measurements from the CATS lidar that operated on International Space Station between 2015 and 2017. The manuscript is a revised version of an earlier submission to ACPD. The primary problem with that submission was that the authors did not address the substantial difference in the CATS lidar calibration from daytime to nighttime that is known

to exist. In the current version the authors have attempted to address this issue by analyzing the daytime and nighttime variations in local time separately. This is not quite a satisfactory way of dealing with the problem. Thus no attempt has been made to disentangle the daytime variations from the large uncertainties (~21%) in total attenuated backscatter coming from the large daytime calibration uncertainties. This may complicate interpretations of the variability with low dust loadings, for instance over North America as well as in the southern hemisphere. Also they continue to use the quality unassured and sparse AERONET nighttime data to assess CATS day/night data quality. In any case the current version has improved particularly in terms of including data from CALIOP as well as providing vertical and seasonal information of the variations and including wind data to interpret the dust loading. They have also revised the manuscript in the light of other comments from the referees on the previous version. The paper is within the scope of Atmospheric Chemistry and Physics and should be useful to the community interested in dust variability using satellite measurements. I have a few minor comments on the current version.

Thank you for supporting our revision of the manuscript and reminding about the remaining uncertainties. We deeply appreciate your input during the last round of review and the following further comments. Regarding the large uncertainty in daytime data, we add "*The large uncertainty of daytime data complicates the assessment of dust loading variability over low-dust regions, such as North America and Southern Hemisphere*" on lines 221-223 of the revised manuscript.

1. There are some significant differences between the CATS plots of seasonal DAOD in Figure 2 and supplemental Figure 3. For example, note the high DAOD values of ~0.2 over the biomass burning areas in southern Africa in JJA in the supplemental Figure 3, which are not there in the corresponding plots in Figure 2. If this is due to the standardization applied to Figure 2 then it should be made clear in the text in section 3.2. I do not see particularly high values of DAOD in the corresponding CALIOP 532nm plots either for 2015-2017 or 2006-2017. The authors have used dust as well as mixture of dust with smoke or marine aerosols, i.e polluted dust and dusty marine in CALIOP terminology. On the other hand they have used only "dust" from CALIOP data. Could this be making any difference in these plots?

Thanks for pointing out the inconsistency between Figure 2 and Supplemental Figure 3. The difference between Figure 2 and Supplemental Figure 3 is mainly due to the different time of day during which the data is aggregated. In Figure 2, corresponding to the morning pass of MISR, data during local time 8 am to noon is averaged to produce the plot. While in Supplemental Figure 3, corresponding to the afternoon pass of CALIOP, data during local time noon to 3 pm is aggregated. The caption of Figure 2 is updated corresponding also to your comment #2.

In term of dust versus dust mixture, we now emphasize throughout the revised manuscript that this study assesses the diurnal variability in dust and dust mixture. This change is reflected in the revised title, "*A Global Analysis of Diurnal Variability in Dust and Dust Mixture Using CATS Observations*", and abstract, among many other sections of the revised manuscript.

The revised abstract reads as "*The current study investigates the diurnal cycle of dust and dust mixture loading across the global tropics, sub-tropics, and mid-latitudes by analyzing aerosol extinction and typing profiles observed by the Cloud-Aerosol Transport System (CATS) lidar aboard the International Space Station. According to the comparison with ground-based and other satellite observations, CATS aerosol and dust/dust mixture loading observations exhibit reasonable quality but significant day-night inconsistency. To account for this day-night inconsistency in CATS data quality, the diurnal variability in dust/dust mixture characteristics is currently examined separately for daytime and nighttime periods. Based on an analysis of variance analytical framework, pronounced diurnal variations in dust/dust mixture loading are generally uncovered during daytime periods and over terrestrial areas. The current*

*study identifies statistically significant diurnal variability in dust/dust mixture loading over key dust sources, including the Bodélé Depression, the West African El Djouf, Rub-al Khali Desert, and western and southern North America, confirming the previous observation-based findings regarding the diurnal cycle of dust emission and underlying meteorological processes in these regions. Significant seasonal dust/dust mixture diurnal variability is identified over the Iraqi and Thar deserts. The identified significant diurnal cycles in dust mixture loading over the vegetated regions in Amazon and tropical southern Africa are hypothesized to be driven by enhanced dust emission due to wildfires*".

We now specifically point out that the current study assesses dust and dust mixture, on lines 133-135 of the revised manuscript, read as "*Note that over regions where dust is less dominant, the total concentration of dust and dust mixture aerosols does not always reflect the abundance of dust particles. Therefore, the currently assessed DAOD represents an upper limit of the dust loading sampled by CATS*".

In addition, we expand the comparison between CALIOP and CATS by separating pure dust (revised Supplemental Figures 3 and 5) from dust and dust mixture (revised Supplemental Figures 4 and 6). This comparison is outlined on lines 170-174 of the revised manuscript, read as "*To be consistent with CATS, CALIOP DAOD are calculated as the vertical integral of extinction over "dust", "polluted dust", and "dusty marine" pixels. Furthermore, the pure DAOD (pDAOD), namely the vertical integral of extinction over "dust" pixels, from both CATS and CALIOP is compared, thereby shedding light on the quantitative difference between pure dust and dust mixture loadings over the less dusty Southern Hemisphere*". The uncertainty from including both dust and dust mixture for the Southern Hemisphere is discussed on lines 242-246 of the revised manuscript, read as "*These observed dust plumes likely originate from dust emission after wildfires in southern Africa (Mahowald et al., 2005; Wagenbrenner, 2017; Wagenbrenner et al., 2013), and/or are likely due to higher amount of biomass burning aerosols mixed with dust. This seasonal dust/dust mixture loading is also present in the pDAOD in both CATS and CALIOP daytime observations over the tropical southeastern Atlantic and southern Africa (Supplemental Figure 3) and doubles when accounting for both dust and dust mixture according to both instruments (Supplemental Figure 4)*".

2. Does Figure 2 include data from both day and night?

Thank you for suggestion on the clarification of Figure 2. Figure 2 only includes data during local morning time. The first sentence of the caption of Figure 2 has been changed to "*Comparison of seasonal average, standardized, local morning DAOD from (a-d) CATS (1064 nm) during local time 8 am to noon and (e-h) MISR (550 nm) around local time 10:30 am in (a, e) December-February (DJF), (b, f) March-May (MAM), (c, g) June-August (JJA), and (d, h) September-November (SON) during 2015-2017*".

3. Lines 119-120: this sentence needs to be rephrased, may be replacing "According to . . .modelling results" by "For example,.." and giving the link to the browse image rather than just giving the NASA center name : https://cats.gsfc.nasa.gov/data/segment_detail/330280/

Thank you for the edit. This sentence has been changed to "*For example, CATS successfully captured a large plume of Saharan dust transporting across the Atlantic Ocean on June 17, 2015 (https://cats.gsfc.nasa.gov/data/segment_detail/330280/)*" on lines 120-123 of the revised manuscript.

4. There are some inconsistencies between the text (lines 321-323) and the plot numbers referred to.

Thank you for correcting the figure numbers. The reference to figures has been updated on lines 328-337 of the revised manuscript.

5. Line 297 and supplemental Figure 5: please specify the CALIOP wavelength used in this Figure —is it still 532 nm?

We change the opening sentence to this paragraph to "*Furthermore, the comparison between dust extinction profiles from CATS (1064 nm) and CALIOP (532 nm) at the Bodélé Depression (Supplemental Figure 7) confirms the existence of "false clear air" near surface in CATS data as identified by Lee et al. (2019)*" on lines 307-309 of the revised manuscript.

6. In Figures 10-13, the number of CATs overpasses within each time window is shown at the bottom of the bars but is barely visible. Please increase the font size or use bold.

Thank you for the suggestion on figure presentation. We have enlarged and boldened the number of CATS overpasses in Figures 10-13. All figures are updated for higher quality and better visualization.